# Are Routine Chest X-rays Necessary following Thoracic Surgery? A Systematic Literature Review and Meta-Analysis

**DOI:** 10.3390/cancers14184361

**Published:** 2022-09-07

**Authors:** Christian Galata, Lorena Cascant Ortolano, Saeed Shafiei, Svetlana Hetjens, Lukas Müller, Roland H. Stauber, Davor Stamenovic, Eric D. Roessner, Ioannis Karampinis

**Affiliations:** 1Division of Thoracic Surgery, Academic Thoracic Center Mainz, University Medical Center Mainz, Johannes Gutenberg University Mainz, 55122 Mainz, Germany; 2Departmental Library, University Medical Center Mainz, Johannes Gutenberg University Mainz, 55122 Mainz, Germany; 3Institute of Medical Statistic and Biomathematics, University Medical Center Mannheim, Medical Faculty Mannheim, Heidelberg University, 68167 Mannheim, Germany; 4Clinic for Diagnostic and Interventional Radiology, University Medical Center Mainz, Johannes Gutenberg University Mainz, 55122 Mainz, Germany; 5Department of Otorhinolaryngology, Head and Neck Surgery, Molecular and Cellular Oncology, University Medical Center Mainz, Johannes Gutenberg University Mainz, 55122 Mainz, Germany

**Keywords:** lung surgery, X-ray, thoracic surgery, lobectomy, perioperative care

## Abstract

**Simple Summary:**

X-rays of the chest have become part of the clinical routine for patients undergoing surgery of the chest. Each of these X-rays exposes the patient and the medical staff to radiation, increasing the treatment costs and the workload. The scientific evidence for performing X-rays after chest surgery (excluding heart surgery) is limited. The purpose of this study was to gather the evidence and analyze it in order to find out how often these X-rays have consequences or lead to a change in patient care. The results of this study could potentially help reduce the number of X-rays that are routinely performed following surgery of the chest.

**Abstract:**

(1) Background: The number of chest X-rays that are performed in the perioperative window of thoracic surgery varies. Many clinics X-ray patients daily, while others only perform X-rays if there are clinical concerns. The purpose of this study was to assess the evidence of perioperative X-rays following thoracic surgery and estimate the clinical value with regard to changes in patient care. (2) Methods: A systematic literature research was conducted up until November 2021. Studies reporting X-ray outcomes in adult patients undergoing general thoracic surgery were included. (3) Results: In total, 11 studies (3841 patients/4784 X-rays) were included. The X-ray resulted in changes in patient care in 488 cases (10.74%). In patients undergoing mediastinoscopic lymphadenectomy or thoracoscopic sympathectomy, postoperative X-ray never led to changes in patient care. (4) Conclusions: There are no data to recommend an X-ray before surgery or to recommend daily X-rays. X-rays immediately after surgery seem to rarely have any consequences. It is probably reasonable to keep requesting X-rays after drain removal since they serve multiple purposes and alter patient care in 7.30% of the cases.

## 1. Introduction

Chest X-ray is probably the only examination that every thoracic surgical patient has been exposed to. The number of X-rays performed in the perioperative setting of elective, general thoracic procedures widely varies. There are surgical units that routinely perform chest X-rays before the upcoming procedure in order to have a “baseline X-ray”, which is then followed by an X-ray in recovery as well as further X-rays on every postoperative day until the drain(s) is out and the patient has been discharged. On the other hand, there are units that only X-ray patients based on their symptoms or when there is a clinical concern [1].

Chest X-rays have several advantages, which have probably led to their widespread in general thoracic surgery. They are available in almost every hospital in the world; the interpretation does not depend on the examiner as strong as the interpretation of ultrasound scans but can be assessed directly by the operating surgeon and confirmed by the radiologist. Additionally, they can be performed on the ward as portable X-rays, and most importantly, they can provide a reasonable answer or at least a hint to most clinical questions which arise during the care of the standard thoracic surgical patient. Furthermore, the X-ray can serve as a way to support the documentation of the postoperative course following general thoracic surgery. 

There are several reasons to request a postoperative X-ray. There are “hard” indications, such as a patient unexpectedly developing respiratory failure after a thoracic procedure or a patient presenting with a very large postoperative air leak, which cannot be explained by the procedure, and less vital indications such as checking the lung expansion or the position of the drain, etc. 

The modernization of general thoracic surgery has changed some previously established indications for requesting X-rays. The standard elective thoracic surgical candidate is not equipped with a central venous line anymore, often does not require an arterial line, and usually requires a single chest drain after the procedure. Furthermore, the typical thoracic surgical patient is extubated immediately after the procedure and can often go directly to the ward after surgery without spending time in a high dependency unit. Hence, the above-mentioned changes in the thoracic surgical practice have eliminated several indications for requesting an X-ray in the immediate postoperative period.

It is therefore more probable that historical reasons and tradition rather than evidence guide our daily clinical routine to perform serial X-rays after thoracic surgery. X-rays expose the patient to radiation, increase our workload, and increase the treatment costs. The purpose of this study was to assess the current evidence on X-rays in the postoperative period following general thoracic surgery with regard to the changes in patient care.

## 2. Materials and Methods

The systematic review was conducted in accordance with the PRISMA guidelines (Preferred Reporting Items for Systematic Reviews and Meta-Analyses) [2]. The objective of the analysis was to determine in patients following non-cardiac thoracic surgery if postoperative X-rays are associated with a significant rate of changes in postoperative care.

All studies reporting X-ray results and associated changes in patient care on patients undergoing thoracic surgery, regardless of the type of the procedure, were included. Studies on pediatric patients (under 18 years of age), animal studies, studies on patients undergoing cardiac surgery, esophageal surgery, and studies on patients in intensive care units were excluded. The reason for exclusion for the latter was that these patients often receive X-rays for several reasons not related to the operation itself. Conference abstracts and unpublished data were also excluded due to poor data availability.

### 2.1. Systematic Literature Research

A comprehensive literature search was performed up until 22 November 2021. The following databases were searched: MEDLINE (PubMed, National Library of Medicine, Bethesda, MD, USA), CINAHL (EBSCOhost, Ipswich, MA, USA), Cochrane Central Register of Controlled Trials (Cochrane, London, UK), Cochrane Database of Systematic Reviews (Wiley), and Web of Science (Clarivate Analytics, Philadelphia, PA, USA). Forward (checking if key papers had been cited) and backward (checking reference lists) citation chasing was performed for key references to ensure that all relevant literature was retrieved. The search strategies used a combination of Medical Subject Heading terms (MESH terms) and free text terms for ‘thoracic surgery’ and ‘X-ray’. The search was not limited by publication type, and there were no restrictions on language. The review was registered in the PROSPERO registry for systematic reviews and meta-analyses (CRD42021287314). Duplicates were removed in EndNote 20 (Clarivate Analytics, London, UK) by the librarian (LCO) following the Bramer Method [3]. The complete search strategy is available as in the Appendix A.

Three reviewers (CG, SS, and IK) independently performed the extraction of the data from the included studies. Data were extracted for the following outcomes: surgical approach, surgical procedure, total number of X-rays performed divided into routine and urgent X-rays, time point of the X-ray, changes in patient care, and X-ray-related costs.

No automatic data extraction tools were used in this study. The findings of the three independent reviewers were controlled for concordance. Disagreements were resolved with discussion and intense analysis of the trials and the data. In order to provide and analyze as much data as possible, the original authors were contacted via email. In case no valid email address was available, a thorough web-based search and contact were attempted. Only reported data were used in the analysis. No assumptions were made for missing/unclear information.

### 2.2. Statistical Analysis

The endpoint of the analysis was if the performed X-ray(s) resulted in a change in patient care. For the purpose of this study and in order to allow future planning of further studies, we defined a 10% change in patient care as significant. The reason we used the 10% margin to define significance was the fact that we defined it as “change in patient care”, any action deviating from doing nothing. A pathological X-ray not requiring any treatment but leading to a further X-ray being requested was defined as a “change in patient care” in this review.

Success rates with 95% CI per study and pooled analysis were shown in the forest plot. The heterogeneity of studies was calculated using the I^2^ index. The random effects model was used for the analysis of pooled data that takes heterogeneity between the studies into account. The weighting of the studies was performed according to the random model of DerSimonian and Laird and is presented in the forest plot. The publication bias analysis is shown in a funnel plot and was examined with Egger’s test.

Three of the included studies allowed patients to be subcategorized into more than one group depending on the indication for the X-ray [1,4,5]. For this reason, the number of X-rays and not the number of patients was used for the statistical analysis. A subgroup analysis was performed to detect the type of incision (VATS or thoracotomy) and if the type of X-ray was associated with the resulting changes in patient management. Subgroup comparison was performed using the Welch–Satterthwaite test. The correlation between a specific type of surgery and a change in patient care was examined according to the Pearson correlation coefficient. Statistical significance was assumed for *p*-values less than 0.05. Statistical analyses were completed using MedCalc software (version 19.6) and SAS software, release 9.4 (Cary, NC, USA).

## 3. Results

The database search provided 5715 references (Figure 1). After removing duplicates, 5159 abstracts were screened for eligibility. Twenty papers were assessed as full-text articles.

The citations of these 20 references were hand-screened in order to identify relevant publications. Hand-screening provided six additional references. Overall, 26 references were assessed as full texts, and 11 were included in the review.

Overall, 11 studies with 3841 patients undergoing 4784 X-rays in the perioperative period were included. Seven studies reported results from X-rays in the immediate postoperative period (PACU-post anesthesia care unit X-ray), which was performed either in recovery, post-anesthesia care unit, or high dependency unit (Table 1). Three studies analyzed the change in patient care after removal of the chest drain (PDR-X-ray: post drain removal X-ray). Four studies reported changes in patient care after X-raying patients for various other indications. As reported above, three studies allowed patients to be subcategorized into more than one group depending on the indication of the X-ray [1,4,5].

The type of incision used in the included studies is presented in Table 1. Overall, two studies reported results following open surgery, four studies after VATS and open surgery, one study after cervical mediastinoscopy, and the other studies reported results following other combinations of the above-mentioned approaches.

The relative frequency of the change in patient management, along with the 95% confidence interval among the included studies, is presented in Figure 2. The three studies reporting results for more than one indication for X-ray are presented twice in the forest plot. The first time each of the three studies appears on the forest plot represents in all three cases the relative frequency of the change in patient management following the PACU X-ray; the second time represents the second indication, as reported in Table 1.

### 3.1. X-ray in the Immediate Postoperative Period (PACU X-ray)

Seven studies analyzing 2284 X-rays reported results from X-rays in the immediate postoperative period. The X-ray led to a change in patient care in 63 patients (2.75%). In mediastinoscopy or thoracoscopic sympathectomy, the probability of the X-ray altering the postoperative management was zero.

### 3.2. X-ray after the Removal of the Chest Drain (PDR X-ray)

In total, 466 X-rays following the removal of the chest drain were available for analysis. The X-ray led to a change in patient care in 34 out of 466 X-rays (7.29%).

### 3.3. Subgroup Analysis

We compared PACU and PDR X-rays with regard to changes in patient management. The relative frequency of PACU X-rays in changing patient management was 2.64% (95% CI: 0.54%; 6.24%). The relative frequency of PDR X-rays in changing patient management was 4.67% (95% CI: 0.18%; 21.18%). PDR X-ray led more often to a change in patient management than the PACU X-ray (*p* < 0.0001).

Furthermore, we examined if there is an association between the type of surgical access and the change in patient care. Patients after minimally invasive surgery (VATS or robotic-assisted surgery) deviated from the normal postoperative care in 0.64% (95% CI: 0.12%; 1.58%) of the cases (Table 2). The relative frequency of thoracotomy patients deviating from the normal postoperative care was 3.60% (95% CI: 0.28%; 10.41%), which was significantly higher than VATS (*p* < 0.0001).

We tried to assess the impact of X-ray-related change in patient care in patients following lung resections. Only one study reported data on patients who had undergone only anatomic lung resections and received daily postoperative X-rays. Daily postoperative X-rays led to a change in patient care in 30.50% of the patients [10].

We assessed if there is a correlation between a specific type of surgery and change in patient care. There was a slight, however non-significant, association between major anatomic lung resections (lobectomy or pneumonectomy) and deviation from the normal postoperative care (Pearson’s coefficient 0.67, *p* = 0.07).

### 3.4. Study Heterogeneity and Publication Bias

The I^2^ index for heterogeneity was 98.18% for all studies. Analysis of heterogeneity of PACU and PDR also showed values above 90% (92.71% and 96.2%, respectively). The subgroup analysis of patients after minimally invasive surgery and thoracotomy showed moderate heterogeneities of 71.91% and 62.56%, respectively. Heterogeneity accounted for all pooled models by the random model. All test results for publication bias were not significant (Egger’s test: *p* > 0.05, Figure 3).

## 4. Discussion

The purpose of this review was to analyze the evidence for performing X-rays in the perioperative window of elective, non-cardiac thoracic surgery. With regard to perioperative imaging, the perioperative window can be divided into four parts: 

The first part would be during the day before the surgery or on the day of the clinical assessment in the outpatient clinic. No studies investigating the benefit of an X-ray during this phase could be found. It is therefore not possible to form an evidence-based recommendation. Patients that are referred with a CT scan or MRI/PET scan do not require an additional chest X-ray routinely in our opinion since it is rather improbable that it will offer additional information. In rare cases when a preoperative X-ray would be helpful in order to compare a potentially pathological postoperative X-ray, the CT scout view can be useful.

The second part is during the period immediately following surgery. This part includes the postoperative X-ray on the post-anesthesia care unit, recovery, high dependency unit, or on the first postoperative day on the ward. We were able to find seven studies assessing this question. By pooling the nearly 2300 X-rays together, we found that in 2.75% of these cases, the X-ray resulted in a change in the treatment course of the patient. As mentioned above, we defined it as “change to the patient care” any deviation from doing nothing, even if that would mean that the patient would only require an additional X-ray. It is therefore reasonable to discuss if the routine X-ray in the immediate postoperative period should be omitted.

The third part comes when the drain can be removed. There are studies supporting the clamping of the drain before removal in order to avoid unnecessary interventions, especially in patients following pneumothorax surgery, where the rate of re-interventions has been estimated at 6.5% [5]. One of the studies in our review reported a change in patient management in 14% of the patients after an X-ray with a clamped drain [4]. There is no sufficient evidence to routinely recommend clamping or not clamping the drain. It is probably reasonable to perform an X-ray if the drain is clamped, but this requires having an X-ray before the drain was clamped in order to assess the differences. If there is no clamping of the drain and the patient is asymptomatic, the drain can probably be removed safely without a previous X-ray. However, there are no studies specifically addressing this question.

The fourth part is after the drain removal. This is a complex part because there is the clinical and the medicolegal point of view. A recently published retrospective study on 433 patients analyzed the rate of re-intervention after post-drain removal X-rays and found that only 3% of the patients required additional interventions, while 33% had an abnormal X-ray [14]. In our review, evidence from three studies with 466 post-drain removal X-rays showed a change in patient care in 7.29% of the cases. It is therefore reasonable to consider performing an X-ray after removal of the drain and before discharging the patient, which can also serve as evidence to confirm the safe discharge status of the patient.

By looking into the subtypes of surgery and the necessity to perform an X-ray, it becomes clear that X-rays following procedures such as cervical mediastinoscopy or thoracoscopic sympathectomy were never associated with changes in patient care. Consequently, routine X-rays in asymptomatic patients following these procedures could probably be omitted.

Looking at the publications of the last ten years and comparing them with the publications of the years before, we can see that in the recent publications, the frequency of an X-ray leading to a change in patient care was 4.35% compared with 18.78% in the years before. This could be potentially explained by the general shift in the practice amongst thoracic surgeons toward adopting a more restrictive approach to obtaining chest imaging.

Finally, the financial implications resulting from the reduction in the number of perioperative X-rays are another significant issue. Graham reported that reducing the number of perioperative portable chest X-rays down to one following thoracic surgery would reduce the cost of perioperative care by USD 725 per patient [7]. Porter estimated a minimum reduction of USD 241.000 in the annual treatment costs of their institution just by eliminating two routine X-rays (the cost of a single X-ray was USD 500) [1]. The cost of a departmental X-ray for an inpatient in our institution varies between EUR 21–54. However, the cost of each X-ray is not only the cost of the materials but also the cost of the radiographer performing the X-ray, the reporting radiologist and the consultant radiologist, and, of course, the cost of the potential prolongation of the length of stay associated with the additional X-rays.

### Limitations

This meta-analysis has several limitations. Despite the high heterogeneity of the I^2^ test in the included studies and the consideration of between-study variance in the statistics, most studies represent retrospective analyses without a control group and with relevant bias. For this reason, risk of bias assessment and sensitivity analysis in order to detect potential reporting bias were not performed. Although the primary endpoint in most studies was the same, so pooling the studies together resulted in a relatively valid outcome, the surgical procedures in each study, the way the primary endpoint was defined and assessed, the surgical approach, and other inherent factors were different. From the clinical point of view, this review does not take the clinical condition of the patient as well as factors such as the level of the air leak, potential changes in the blood tests, etc., into consideration. These data could not be retrieved from the available studies and should certainly be part of future research on defining risk factors that could increase the probability of an X-ray determining the change in patient care.

## 5. Conclusions

Performing serial X-rays in the perioperative window of general thoracic surgery has several disadvantages. It is associated with increased workload, increased hospital costs, and, most importantly, exposure of the individual patient, further patients, and medical staff to radiation. There is no strong evidence to support performing any X-ray apart from the X-ray after having removed the pleural drain. Despite the several significant limitations, this review shows that the probability of a perioperative X-ray having a relevant consequence is rather low. It is therefore reasonable to question each indication for an X-ray on an asymptomatic patient and think about the potential consequence that the X-ray would result in.

## Figures and Tables

**Figure 1 cancers-14-04361-f001:**
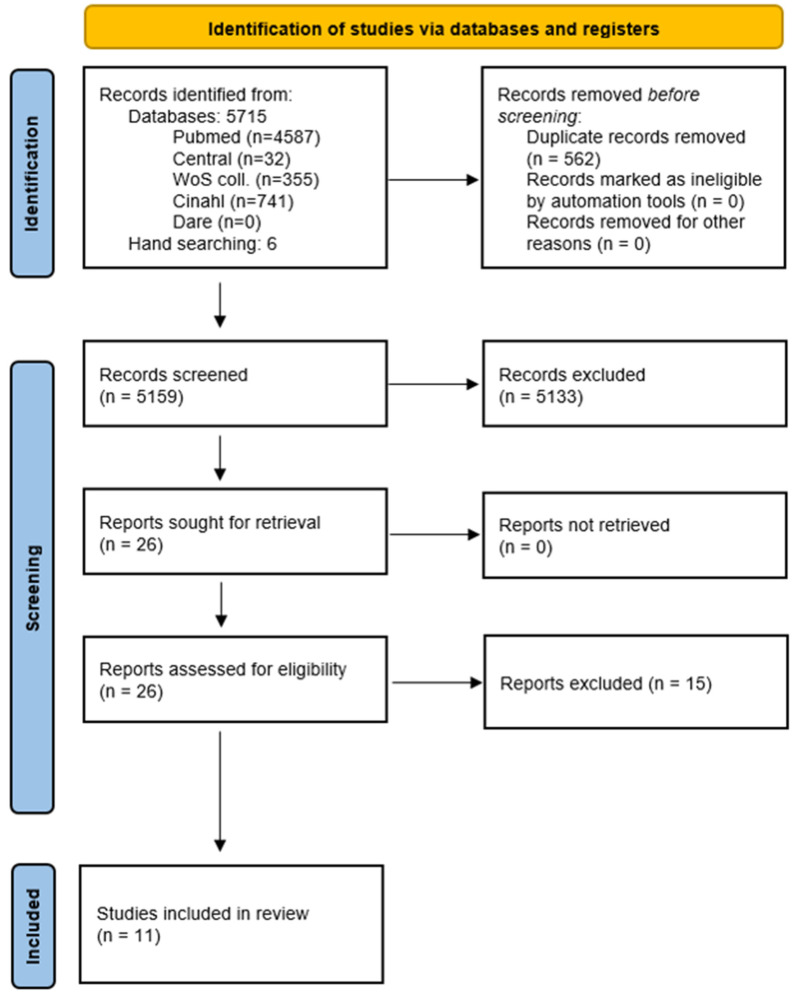
PRISMA flowchart.

**Figure 2 cancers-14-04361-f002:**
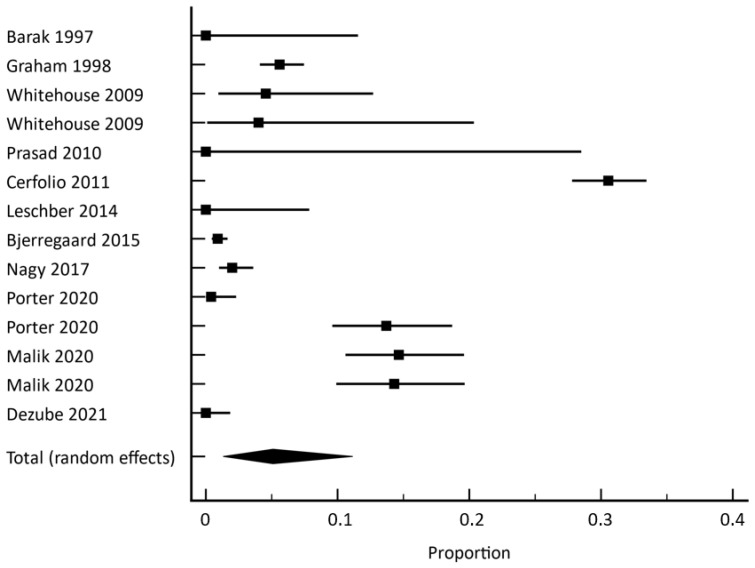
Forest plot of the meta-analysis. The relative frequency of a change in patient care for all included studies is presented with a 95% confidence interval. The diamond shows the result of the pooled analysis of the random effect model [1,4,6,7,8,9,10,11,12].

**Figure 3 cancers-14-04361-f003:**
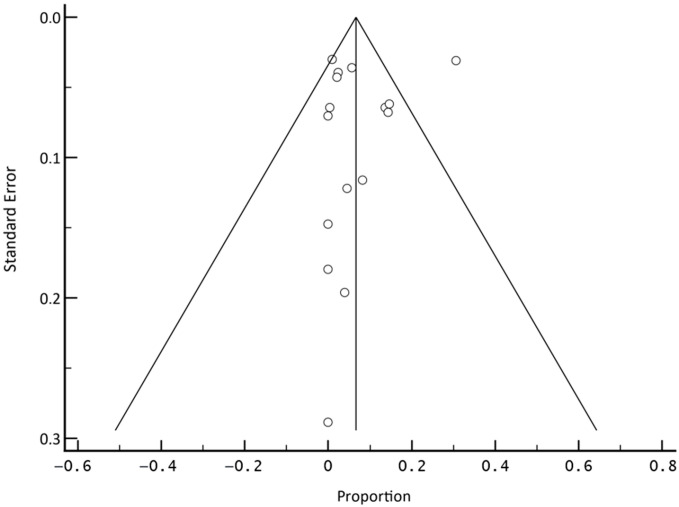
Funnel plot of the meta-analysis of published studies. Each plotted point represents the standard error and relative frequency for one study. The triangle represents the region where 95% of the data points are located. The vertical line represents the average relative frequency.

**Table 1 cancers-14-04361-t001:** Included studies.

N	Author	Year	Study Type	X-ray TP	Surg. Approach
1	Barak [6]	1997	Prospective	PACU	Open
2	Graham [7]	1998	Prospective	Various	Open
4	Whitehouse [4]	2009	Prospective	PACU + PDR	VATS/Open
5	Prasad [8]	2010	Prospective	Symptom-based	VATS
6	Cerfolio [9]	2011	Retrospective	Daily	VATS/Open
8	Leschber [10]	2014	RCT	PACU	Mediastinoscopy
9	Bjerregaard [11]	2015	Retrospective	PACU	VATS
10	Nagy [12]	2017	Prospective	PACU	VATS/Open
11	Porter [1]	2020	Retrospective	PACU + PDR	VATS/RATS/Open
12	Malik [5]	2020	Prospective	PACU/clamping	VATS/Open
13	Dezube [13]	2021	Retrospective	PDR	VATS/RATS

TP: time-point; RCT: prospective randomized trial; PACU: post-anesthesia care unit; PDR: post-drain removal X-ray; clamping: X-ray after clamping the drain; VATS: video-assisted thoracoscopic surgery; RATS: robotic-assisted thoracoscopic surgery.

**Table 2 cancers-14-04361-t002:** Surgical procedures and change in patient care.

N	Author	Patients/X-rays	Surgical Procedure	Change in Pat. Care
1	Barak [6]	30	Not reported	0/30
2	Graham [7]	99/769	P: 12 L: 37 W: 33 O:18	43/769
4	Whitehouse [4]	74/91	P: 1 L: 11 W: 24 O: 38	PACU 3/66 PDR 1/25
5	Prasad [9]	322/11	Sympathectomy	0/11
6	Cerfolio [10]	1037/1037	L: 609 S: 146 W: 282	317/1037
8	Leschber [12]	93/45	Cervical mediastinoscopy	0/45
9	Bjerregaard [13]	1002/1097	P + L: 344 W: 619 O: 134	10/1097
10	Nagy [14]	546/546	P + L: 191 O: 355	11/546
11	Porter [1]	241/482	P: 1 L: 80 W: 71 O: 89	PACU 1/241 PDR 33/241
12	Malik [5]	197/476	L: 48 W: 118 O: 131	PACU 38/259 Clamping 33/217
13	Dezube [13]	200/200	L: 59 S: 24 W: 117	0/200

P: pneumonectomy; L: lobectomy; S: segmentectomy; W: wedge resection; O: other procedure; PACU: post-anesthesia care unit; PDR: post drain removal X-ray.

## Data Availability

Not applicable.

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
