# Peer review of "Are Routine Chest X-rays Necessary following Thoracic Surgery? A Systematic Literature Review and Meta-Analysis"

_cancers, 2022, doi:10.3390/cancers14184361_

Round 1

Reviewer 1 Report

I applaud the authors for exploring this topic which, while often underappreciated, carries significant clinical relevance.  CXRs are often perceived as innocuous and inexpensive tests that can be obtained with impunity.  However, there has been increasing evidence showing significant cost burden associated with the routine use of CXRs, while demonstrating no clear clinical benefit of this practice in asymptomatic patients. 

This is a well-designed study with a clear methodology and inclusion criteria.  However, there are a few issues to address before this can be considered for publication.

I question whether some of your included studies actually meet your inclusion criteria.

1.  Barak – this study was not centered around thoracic surgical patients.  The indication for CXR was predominantly for CVC/PAC insertion or mechanical ventilation, with fewer than 25% being performed for thoracotomy/thoracoscopy.  Although the authors report no changes in management with thoracotomy patients, it’s not clear whether another indication for CXR existed in these patients (concomitant CVC/PAC or mechanical ventilation).  This would confound the result.  I would not recommend including this study in your review.

2.  Palesty – this study included trauma and general surgical patients, not thoracic surgery.  Given that the purpose of this study is to assess CXR usage in thoracic surgery patients, this study inappropriate to include in your review.

3.  Prasad – I do not have access to the full article.  However, based on the abstract, the main objective of this study is to assess outcomes of sympathectomy on hyperhidrosis and not to specifically evaluate CXR use after this procedure.  I would re-consider the appropriateness of this study for your review.

4.  Knudsen – 69% of the patients in this study were <18 y/o.  Your study design indicated that pediatric patients were excluded.  You will need to eliminate the pediatric patients in this study, or exclude it altogether. 

You mention that CXRs increase treatment costs in the introduction but do not evaluate or discuss the cost burden incurred with excessive use of CXRs.  Since one of the major driving forces for eliminating unnecessary CXRs is cost reduction and resource preservation, I would recommend expanding your discussion to include the potential cost savings.  Several of the studies you cited reported a cost analysis.  It would be interesting to see how much could be saved annually by eliminating unnecessary CXRs.  A few examples taken from your cited references:

1.  Graham – cost savings of 95% if routine CXRs were eliminated

2.  Porter – cost savings of $241,000 per year if routine CXR eliminated

3.  Zukowski – cost savings of $180,000 (87% reduction) if routine PDR-CXR eliminated

Author Response

Point 1: You reported that Barak’s paper only had a small number of patients undergoing thoracic surgery and many other patients undergoing x-rays for other indications, like central lines, which is absolutely correct. We were not able to include all patients from Barak’s paper in the analysis but only the patients undergoing thoracotomy, not even the patients who underwent thoracoscopy, because we were not able to extract the necessary data. It is, of course, possible (we are not sure if this is what you meant with your comment) that -at least- in some of the patients who underwent thoracotomy the x-ray was requested for checking the CVC line but this will probably apply to a lot of patients that are included in this study and it is probably something that we will never be able to rule out in a retrospective study.

Points 2+3: You are absolutely right; all these patients received a chest x-ray following drain removal for trauma and none of them underwent a formal thoracic procedure. From our point of view, this question applies also to patients that underwent thoracoscopic sympathectomy; We personally don’t consider a VATS sympathectomy to be significantly more of a thoracic surgical procedure than a trauma drain and it is certainly not comparably to a procedure like segmentectomy or lobectomy, this is why we performed the subgroup analysis.

However, the vast majority of thoracic trauma is treated just with a pleural drain. Therefore, we will probably never have solid evidence on x-rays following drain removal after formal thoracic surgery for trauma just because this type of surgery is very rare and the patient cohort very inhomogeneous. From our point of view it would be incorrect/unfair to leave out the only piece of evidence that we will probably be able to collect about chest trauma and x-rays.

Point 3: This study has as a primary endpoint the “prospective evaluation of the thoracoscopic sympathectomies”. However, in the “results” session they clearly report the data that were required to include the study in the analysis.

Point 4: Thank you for raising this issue, which is something that troubled us during the screening process for this study. We had initially set a filter to rule out pediatric studies. This is the only study of this type that we came across during the review process. The study does not report data in a way that would allow us to extract information only for adult patients. On the one side, this study reports data on patients that are underage in 69% of the cases. On the other side, this is the only study on chest wall deformities reporting x-ray results, the median age of the patients is 16 years old, which is not that far from 18 years old (if we consider 18 the limit for naming someone an adult) and this particular type of chest wall deformities is often treated by thoracic surgeons and not by pediatric surgeons. We fully agree with you that it is worthwhile to consider excluding this paper and we had discussed this issue within the team before including the reference. However, every time we exclude a reference, we intentionally conceal information and in this case, we are not absolutely certain that this piece of information should be left out. We would like to have this question forwarded to the other reviewers/editorial team, if there is an agreement that this paper should be taken out, we are happy to repeat the meta-analysis and re-write the paper without this reference.

Point 5: An additional paragraph has been added in the “discussion” reporting the changes/savings resulting from reducing the number of requested x-rays.

Reviewer 2 Report

The authors reported the utility of perioperative X-rays in thoracic surgery as a meta-analysis. I think this manuscript is very interesting and useful for readers in this field.

Comments

1. Reference 8 is not related to thoracic surgery, but tube thoracostomy. I think this report should not be included in this study.

2. Table 1: clamping: x-rax -> x-ray

3. In Table 1, the reported nations should be added.

Author Response

Point 1: Thank you for the comment, which was also raised by reviewer 1. As reported above, it is well-known that the vast majority of thoracic trauma is treated just with a pleural drain. It is therefore unlikely that we will ever acquire solid evidence on x-rays following drain removal after formal thoracic surgery for trauma. We had discussed this issue with the other authors before deciding to include this study in the analysis and decided that it would be incorrect/unfair to leave out the only piece of evidence that we will probably be able to collect about chest trauma and x-rays.

Point 2: Thank you for noticing, we have changed it.

Point 3: What exactly do you mean by adding the nations? Do you wish to have the countries where the studies were conducted in? If yes, we are happy to add it.

Reviewer 3 Report

Manuscript needs major revision

Abstract should be more  concise  and  clear  with clear conclusion for this study, based on the meta analysis but to be focused

If Authors say  we need prospective study to make conclusion, that is mean this study not designed good enough because you got no answer

This manuscript needs focus. Authors can study disease or to  study trauma  and needs x-ray  for consideration after  surgery

If Authors choose disease for example tumors in the thorax, the Authors need to elaborate  more on  the indication for the thoracic surgery and when never can not be performed  for example if size is oved  10 cm  or separate  tumors  and  lymphoma, plasmocytoma, sarcoma and explain when surgery is needed and when is prohibited base on histopathology of the disease or if you choose Sympathectomy or trauma, conclusion should be on that subject not in general

Discuss x-ray prior or post operative necessities based on the disease  not in  general

Author Response

Point 1: The abstract has been revised and the conclusion has been written in a more concise way.

Point 2: You have reported that our manuscript needs to focus either on disease or trauma. The focus of the manuscript or in other words, the endpoint of the analysis is the change in patient care resulting from postoperative x-rays after thoracic surgery. There is only one study reporting results following trauma and there are several studies reporting results following specific procedures (different types of lung resections or other procedures as reported in the subgroup analysis). Not all studies report data on the underlying disease, for which the patients underwent thoracic surgery (cancer, emphysema or others). Furthermore, such data cannot always be pooled together in a formal analysis. We totally agree, the ideal question for the trial would be: “do patients after lobectomy for primary lung cancer benefit from omitting the x-ray in the recovery unit?” Unfortunately, there are no data to answer such questions. On the other hand, this is one of the reasons that such reviews-analyses are beneficial- they bring questions like this and gaps in evidence in the front. These are unfortunately questions that can be answered with prospective trials and as unfulfilling as it sounds, this has to be at least part of our recommendation/closure statement. It is an unsafe practice and incorrect interpretation of evidence to form recommendations based on the results of -mostly- retrospective studies.

Point 3: As reported above, we would also like to be able to create such precise subgroups in order to answer questions like that and pool data like tumor size together but we are afraid that none of the existing studies report such results. This is the reason why we did the subgroup analysis in the first line, in order to be able to answer questions on specific subtypes of surgery, which we were unfortunately only able to do partially (see Subgroup analysis and discussion accordingly)

Point 4: As reported in the subgroup analysis, based on the available data we were only able to pool specific types of lung resection together and were not able to find any significant correlation between lobectomy or pneumonectomy and x-rays. Unfortunately, apart from sympathectomies and mediastinoscopies, we cannot form any other recommendations because there are no data to support such recommendations.

Round 2

Reviewer 1 Report

Thank you for your responses to the previous inquiries.  Although you acknowledge the limitations and suggestions for revision, the concerns raised during the initial review have remained unaddressed.  

The significant heterogeneity of the studies selected for inclusion renders your review completely inconclusive.  For instance, if you are going to include trauma patients (Palesty et al.) with the rational that non-operative trauma managed with a chest tube is equivalent to a patient who underwent a thoracic surgical operation, then why not include the multitude of other similar studies that have been published in trauma patients?  To name a few:

1.  Kong V, Oosthuizen G, Clarke D. What is the yield of routine chest radiography following tube thoracostomy for trauma? Injury. 2015; 41(1):45-8.

2.  Pacanowski JP, Waack ML, Daley BJ, Hunter KS, Clinton R, Diamond DL, Enderson BL. Is routine roentgenography needed after closed tube thoracostomy removal? J Trauma. 2000; 48(4):684-8.

3.  Beattie G, Cohan CM, Chomsky-Higgens K, Tang A, Senekigian L, Victorino GP. Is a chest radiograph after thoracostomy tube removal necessary? A cost-effective analysis. Injury. 2020; 51(11):2493-9.

The same argument can be made for chest tubes placed for medical (non-surgical) indications, such as pleural effusions and spontaneous pneumothorax.  For example:  Diaz R, Patel KB, Shekar SP, Almeida P, Mehta JP. Are Chest Radiographs Routinely Indicated After Chest Tube Removal Following Non-Surgical Placement? Cureus. 2020; 12(3):e7339. How are these different than managing a hemothorax or pneumothorax due to trauma? 

If you are going to include a study specifically evaluating sympathectomies, then why not include esophagectomies?  Kingma BF, Marges OM, Van Hillegersberg R, Ruurda JP. Routine chest X-rays after the removal of chest tubes are not necessary following esophagectomy. J Thorac Dis. 2019;11(Suppl 5):S799-804

Regarding pediatric patients, it is imprecise to assert that “16 years old is not that far from 18.”  If you are specifically excluding pediatric patients, then a strict cutoff must be established, whether based on age or by the treating service.  As you correctly point out, the main difference between peds and adults is the primary managing team (peds vs thoracic surgery).  Adding pediatric studies to your review will significantly increase the heterogeneity of this study.  If you intend to include pediatric patients, there are several published studies to consider for your review:

1.  Johnson B, Rylander M, Beres AL. Do X-rays after chest tube removal change patient management? J Pediatr Surg. 2017; 52(5):813-5.

2.  McGrath E, Ranstrom L, Lajoeie D, McGlynn L, Mooney D. Is a Chest Radiograph Required After Removal of Chest Tubes in Children? J. Pediatr Health Care. 2017; 31(5):588-93.

3.  Pacharn P, Heller DN, Kammen BF, Bryce TJ, Reddy MV, Bailey RA, Brasch RC. Are chest radiographs routinely necessary following thoracostomy tube removal? Pediatr Radiol. 2002; 32:138-42.

The bottom line is that this review has selected a very small sample of 13 studies to evaluate the necessity of routine CXRs after thoracic surgery, and yet these few studies are far from representative of the typical thoracic surgery population.  Absolutely no conclusion on thoracic surgery patients can be drawn from this manuscript.  I agree that there is a paucity of studies that have reported solely on adult non-cardiac, thoracic surgery patients, which makes a comprehensive review quite challenging.  However, unless you plan to broaden your inclusion criteria to include trauma, peds and other non-thoracic surgical indications for chest tubes, this review lacks focus and does not adhere to your methodology.  I continue to find this topic very interesting and worthy of further investigation, but strongly feel that the selection of studies included in your meta-analysis needs to be re-evaluated. 

Author Response

Dear Colleague,

Thank you again for the time and effort that you invested in our manuscript, we truly appreciate it.

The study on trauma patients and the study on underage patients have been removed from the paper.

We are not sure if you have recommended removing the study with the thoracoscopic sympathectomies (Prasad, Surg Endosc (2010) 24:1952–1957). The reason why we included this study and not other studies on esophageal resections is that we had excluded esophagectomies during the literature research (not with a filter but during the hand screening process). Esophageal surgery is not part of the surgical curriculum of thoracic surgeons in our country anymore and as far as we are concerned in many countries in the meanwhile and that is the reason that we did not think about mentioning the exclusion. Thank you for noticing that. We have added a comment in the M&M section accordingly.

We hope that our changes meet your expectations.

Kind regards,

Christian Galata, Davor Stamenovic & Ioannis Karampinis

Round 3

Reviewer 1 Report

I am still quite puzzled why esophageal surgery is excluded.  Although pneumothorax may not be a common finding after esophageal surgery, there are other relevant reasons for obtaining CXR in these patients that are directly related to the operation itself, including pleural effusion, subcutaneous emphysema and chylothorax.  Your argument that there are other reasons unrelated to surgery itself for obtaining CXR in these patients can apply to any of the patients in these studies. As mentioned in an earlier review, the majority of patients in the study by Barak et al. had a CVC/PAC and/or were intubated, which certainly confounds the indications for CXR among those who underwent thoracic surgery.  There is no logical reason why the paper by Kingma et al. should be excluded.    

Regarding the sympathectomy paper, I feel that my point was not made clear.  My objection to that paper is that the aim of the article was NOT on chest tube management, but rather the outcomes of the operation itself.  Perhaps they provide enough information to extract the relevant data.  Unfortunately, I am unable to access the full article at my institution to evaluate its relevance to your review.  If you are able to provide the full text article to me, it would be appreciated.

Also, you cite Zukowski et al. in your discussion but this is not included in your actual review.  This article is extremely relevant to your review and I am curious as to why it is not included in your analysis. 

I disagree with the assertion that frequency of CXR leading to change in patient care decreasing by 18% to 4% is explained by the introduction of electronic suction.  There is nothing in these articles to support that statement.  As far as I can tell, none of the articles explicitly state that they use a digital drain system.  In the US, these digital drains have not yet been widely adopted and it would be a huge leap to consider that the main reason we see a reduction in changes in patient care.  A more likely reason is the general shift in practice amongst thoracic surgeons towards adopting a more restrictive approach to obtaining chest imaging. 

For future revisions, please specify the major changes that were made, such as the addition or removal of particular data points and full studies to or from your analysis.  Even with the highlighting, it is quite difficult to ascertain the changes made.

Author Response

Please note that our reply has been uploaded as a doc. file through the available funnel